



# Atmospheric convergence zones stemming from large-scale mixing

Gabriel M. P. Perez[1], Pier Luigi Vidale[1,2], Nicholas P. Klingaman[1,2], and Thomas C. M. Martin[3]

[1]Department of Meteorology, University of Reading, Reading, United Kingdom
[2]National Centre for Atmospheric Science, Reading, United Kingdom
[3]Department of Atmospheric Sciences, University of São Paulo, São Paulo, Brazil

**Correspondence:** Gabriel M. P. Perez (gabrielmpp@protonmail.com)

**Abstract.**

Organised cloud bands are important features of tropical and subtropical rainfall. These structures are often regarded as convergence zones, alluding to an association with coherent atmospheric flow. However, the flow kinematics is not usually taken into account in classification methods for this type of event, as large-scale lines are rarely evident in instantaneous diagnostics such as Eulerian convergence. Instead, existing convergence zone definitions rely on heuristic rules of shape, duration and size of cloudiness fields. Here we investigate the role of large-scale turbulence in shaping atmospheric moisture in South America. We employ the Finite-Time Lyapunov Exponent (FTLE), a metric of deformation among neighboring trajectories, to define convergence zones as attracting Lagrangian Coherent Structures (LCSs). Attracting LCSs frequent tropical and subtropical South America, with climatologies consistent with the South Atlantic Convergence Zone (SACZ), the South American Low-level Jet (SALLJ) and the Intertropical Convergence Zone (ITCZ). In regions under the direct influence of the ITCZ and the SACZ, rainfall is significantly positively correlated with large-scale mixing measured by the FTLE. Attracting LCSs in South and Southeast Brazil are associated with significant positive rainfall and moisture flux anomalies. Geopotential height composites suggest that the occurrence of attracting LCSs in these regions is related with teleconnection mechanisms such as the Pacific-South Atlantic. We believe that this kinematical approach can be used as an alternative to region-specific convergence zone classification algorithms; it may help advance the understanding of underlying mechanisms of tropical and subtropical rain bands and their role in the hydrological cycle.

## 1 Introduction

Large-scale organised zones of cloudiness and rainfall stand out in tropical and subtropical weather, which is otherwise dominated by non-organised convection. These cloud bands were initially identified in the equatorial belt (Alpert, 1945) and associated with the interaction of inter-hemispheric air masses along the easterlies (Simpson, 1947); i.e., the Intertropical Convergence Zone (ITCZ). Despite these historical associations with coherent trajectories, convergence zones have been more frequently identified by heuristic rules applied to satellite imagery or cloudiness/rainfall data (Barros et al., 2000; Van Der Wiel





et al., 2015; Ambrizzi and Ferraz, 2015; Vindel et al., 2020). These *a posteriori* approaches require detailed previous knowl-
edge of the spatio-temporal characteristics of convergence zones in specific locations. Therefore, they do not provide a general
definition for these events.

In other studies, convergence zones were characterized using the divergence of instantaneous or average velocity fields
(Berry and Reeder, 2014; Weller et al., 2017). These approaches are in principle more general. However, because Eulerian
metrics such as divergence reveal instantaneous features in their immediate neighborhoods, they respond strongly to local
processes such as convection. More generally, in unsteady flows, Eulerian features do not reveal the underlying structures of
tracer mixing such as air mass interfaces (Boffetta et al., 2001; d'Ovidio et al., 2009). Rather, from the kinematics point of
view, flow structures shaping tracer evolution are more suitably inspected under Lagrangian frameworks (Ottino, 1989; Pierre-
humbert, 1991; Bowman, 1999; Haller and Yuan, 2000). By offering temporally integrated trajectory information, Lagrangian
diagnostics synthesize pathline features that determine atmospheric transport.

In this study, we investigate attracting coherent structures arising from large-scale mixing in South America and their rela-
tionship with rainfall and water vapour. We propose that such structures are skeletons of atmospheric convergence zones and
propose an identification criterion that can be applied to reanalyses and model data (Sections 2 and 3). In Section 4, we dis-
cuss the physical interpretation of the quantities involved. In Section 5, we discuss the methodology applied to a recent South
Atlantic Convergence Zone (SACZ) event. Finally, we present their impacts on rainfall and moisture fluxes in South America
(Section 6) and associate them with teleconnections (Section 7).

## 1.1 Mixing and Lagrangian coherent structures in the atmosphere

Regularly distributed tracers in turbulent flows evolve into complex and irregular patterns. This process of mixing is charac-
terized by the stretching and folding of material lines advected in such flow (Ottino, 1989). Mixing can be observed in the
atmosphere on synoptic time scales even in relatively simple flows (Welander, 1955). We consider convergence zones, from
the point of view of mixing, as coherent structures associated with strong attraction of trajectories.

The Finite-Time Lyapunov Exponent (FTLE) is a convenient tool to visualize underlying structures of flow mixing. It is
defined as the average separation rate among neighboring trajectories in a fixed propagation time interval. Ridges of the FTLE
identify Lagrangian Coherent Structures (LCSs) (Haller, 2001; Shadden et al., 2005), structures by which advected passive
tracer is strongly attracted or repelled within the time intervals of interest. While this is an active area of research and more
rigorous methods to investigate Lagrangian coherence exist (Haller and Beron-Vera, 2012; Farazmand et al., 2014), the FTLE
has been employed to investigate features of atmospheric and oceanic transport.

Pioneering studies by Pierrehumbert (1991) and Pierrehumbert and Yang (1993) applied the FTLE to investigate large-scale
atmospheric mixing and tropics-extratropics transport barriers. Shepherd et al. (2000) computed probability density functions
of the FTLE to investigate chaotic advection in the stratosphere. Rutherford et al. (2011) and Guo et al. (2016) employed
the FTLE to visualize flow features in tropical cyclones. Garaboa-Paz et al. (2015) and Garaboa-Paz et al. (2017) suggested
that FTLE ridges are closely linked to atmospheric rivers in boreal winter, when advection shapes the spatial distribution of
water vapour. The criterion for convergence zones proposed here is closely related to the framework proposed by Garaboa-Paz

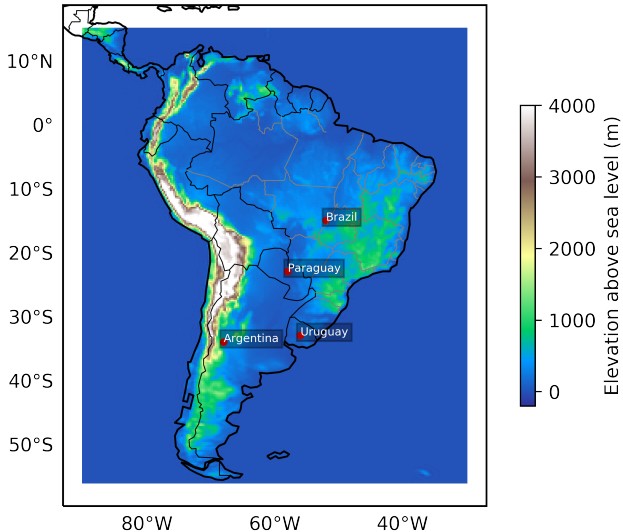

**Figure 1.** Elevation above the sea level. Derived from ECMWF's ERA5 surface geopotential.

et al. (2015) for atmospheric rivers. We leverage the framework of LCSs to investigate the underlying flow features organising rainfall and moisture accumulation in South America.

## 1.2 Aspects of the moisture transport in South America

The largest portion of the South American continent is located in tropical and subtropical latitudes (Figure 1). Along the western coast, the Andes mountain range extends across a considerable latitudinal interval as the dominant topographical feature. Its high altitudes pose a barrier to low and mid-tropospheric zonal flow. This barrier to the zonal flow deflects the climatological sea-to-land easterly flow as it enters the continent from the equatorial Atlantic, forming a meridional channel of north-to-south moisture transport between 850 and 700 hPa (Gimeno et al., 2016) that supplies moisture for rainfall in populated areas in southeastern and southern South America (Zemp et al., 2014). When intensified, this north-to-south moisture flux characterizes the South American Low-level Jet (SALLJ, Vera et al. 2006). Occasionally, the SALLJ resembles atmospheric rivers (Arraut et al., 2012; Poveda et al., 2014), filaments of intense moisture associated with the motion of extratropical cyclones (Dacre et al., 2015).

During austral summer, increased sensible heat flux from the land surface and latent heat released through Amazonian convection, combined with the southward seasonal shift of the ITCZ, intensify the sea-to-land moisture transport, characterizing the wet phase of the South American Monsoon System (SAMS, Marengo et al. 2012). The SAMS can be illustrated by the contrast between rainfall and moisture flux climatologies in summer and winter (Figure 2). In summer, there is increased oceanic moist air input to the continent, associated with the tropical Atlantic easterlies and the South Atlantic Subtropical High (SASH). The northeasterly moisture flux is deflected southeast after crossing the Amazon, supplying moisture for southeastern

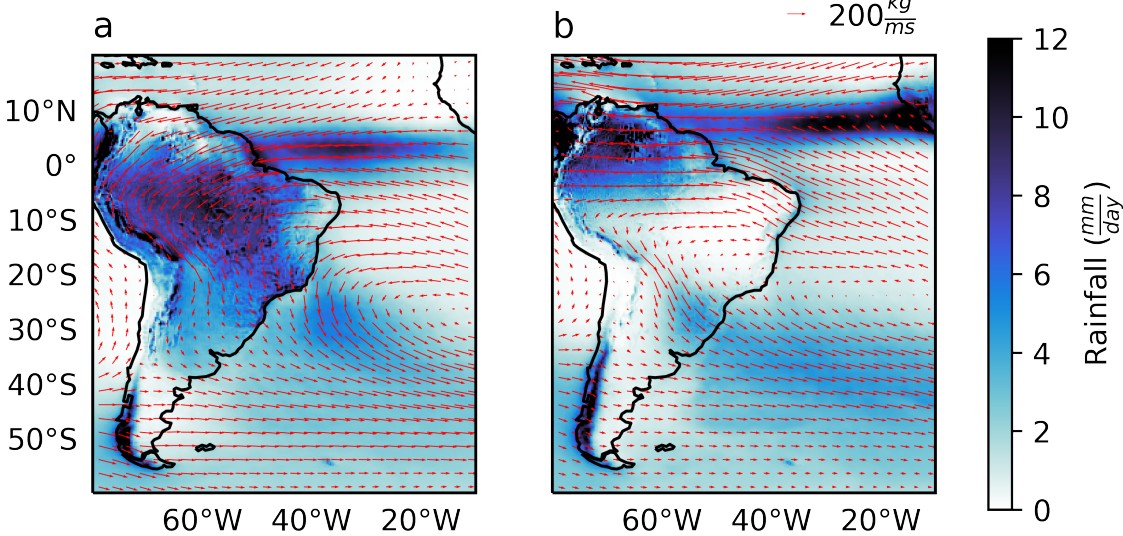

**Figure 2.** Mean rainfall and vertically integrated moisture flux (vectors) in DJF (a) and JJA (b). Averages were computed from ECMWF's ERA5 from 1980 to 2009.

and southern South America (Zemp et al., 2014). This higher moisture input in summer coincides with higher rainfall, particularly in Central and Southeast Brazil. In winter, the angle of the easterlies in the Atlantic tilts northwards and the moisture flux from the SASH weakens, such that the moisture flux and rainfall are stronger in northernmost South America and weaker in most other parts of the continent. These observations point to the sensitivity of rainfall in central and southern South America

to disruptions in the climatological SAMS moisture transport.

Extratropical cyclones are central to the synoptic scale variability of the South American climate. These structures originate most frequently in cyclogenesis regions spanning from South Argentina to the coast of Southeast Brazil (Crespo et al., 2020). These cyclones have an average life cycle of three days (Mendes et al., 2010); their associated cold fronts cause cold incursions (Lanfredi and De Camargo, 2018) and rainfall over the continent (Lenters and Cook, 1999; Vera et al., 2002). In the wet

season of the SAMS, the interaction between tropical sources of heat and moisture with extratropical cyclones resonate at submonthly scales often manifested as diagonally oriented (northwest-southeast) cloud bands (Nieto and Chao, 2013; Raupp and Silva Dias, 2010). The diagonal aspect of these cloud bands is a characteristic feature of the SACZ; it has been attributed to the deformation of low-level vorticity centres by equatorward Rossby waves originating from circulation anomalies in the Pacific (Van Der Wiel et al., 2015). This relationship renders circulation and rainfall in the SACZ region sensitive to the

Pacific-South American (PSA) teleconnection patterns (Mo and Paegle, 2001).

Defining and identifying these mechanisms of submonthly rainfall variability and large-scale moisture transport is, thus, key to understanding the variability of the continental climate. Currently, most identification algorithms for convergence zones





focus on the SACZ and rely on rules of shape, duration and intensity of cloud bands usually given by an Empirical Orthogonal Function (EOF) analysis of outgoing longwave radiation or rainfall (Barros et al., 2000; Jorgetti et al., 2014; Van Der Wiel et al., 2015; Ambrizzi and Ferraz, 2015). A drawback of cloudiness-based approaches is that they lack the means for attributing contributions from processes at different scales to a single cloud band. For example, cloudiness originating from local convection adjacent to cloudiness originating from large-scale flow coherence would appear to belong to the same structure from a satellite image. Furthermore, describing a physical phenomenon with a single EOF is problematic, as EOFs do not necessarily individually correspond to dynamical modes and can produce patterns with little connection to physical processes (Dommenget and Latif, 2002; Monahan et al., 2009; Fulton and Hegerl, 2019). This study is motivated by the need to objectively link convergence zones to atmospheric flow features, which cannot be done using existing definitions.

## 2 Mathematical framework

### 2.1 Vertically scaled horizontal moisture flux

Water vapour in the atmosphere concentrates close to its source at the Earth's surface. Thus, it is natural to analyse features of moisture transport using low-level flow. Weller et al. (2017) employed an Eulerian metric to identify convergence lines at 950 hPa over the Pacific Ocean and Australian landmass. Garaboa-Paz et al. (2017) investigated atmospheric rivers at 850 hPa. While water vapour is concentrated at lower levels, selecting a particular level becomes problematic near topography, such as the Andes, for two reasons: (a) topography often crosses lower tropospheric pressure levels; and (b) it causes the level of maximum moisture transport to rise in its vicinity (Insel et al., 2010). The SALLJ, for example, transports substantial amounts of water vapour and flows parallel to the Andes between 850 and 700 hPa (Gimeno et al., 2016).

We employ a horizontal flow $\boldsymbol{V}_{\rho_v}$ derived from a vertical scaling of the horizontal momentum $\boldsymbol{V}_H$ that takes into account the vertical distribution of water vapour density ($\rho_v$). The scaling divides the vertically integrated moisture flux (VIMF) by the total column water vapour (Eq. 1). Physically, $\boldsymbol{V}_{\rho_v}$ is the average flow by which the total column water vapour is transported. A similar weighting was employed by Garaboa-Paz et al. (2015) to identify atmospheric rivers and by Ruiz-Vasquez et al. (2020) to investigate sources and sinks of water vapour in South America. It is important to notice, however, that strong vertical shear across heights of high moisture concentration may render $\boldsymbol{V}_{\rho_v}$ not representative of the actual horizontal pathways of moisture, obscuring the interpretation of attracting structures in this flow.

$$\boldsymbol{V}_{\rho_v} = \frac{\int_0^\infty \rho_v \boldsymbol{V}_H dz}{\int_0^\infty \rho_v dz} [m/s] \tag{1}$$

### 2.2 Finite-time Lyapunov exponent

The FTLE measures the average deformation rate among initially close trajectories after a characteristic advection time $\Delta t = t_1 - t_0$, where $t_1$ and $t_0$ are respectively the times of arrival and departure of these trajectories. The flow-map $F_{t_0}^{t_1}$ links the

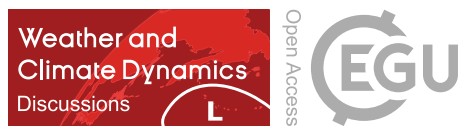

**1. Input data**

| Vertically integrated moisture flux | Total column water vapour |
|---|---|

1980
2009

**2. Scaling**

Eq. 1

Vertically scaled velocity

1980
2009

**3. Flow-map**

For sliding windows of 2 days; Step by 6 hours

Eq. 3 with second-order scheme "SETTLS"

*Python function: LCS.trajectory.parcel_propagation*

Departure points from back-trajectories (backwards flow-map)

Example 2-day back-trajectories arriving in different neighborhoods at the same time

**4. FTLE**

Eq. 5

*Python function: LCS.LCS.compute_deftensor*

Cauchy-Green strain tensor

Eq. 4

*Python class: LCS.LCS.LCS*

FTLE

FTLE for the 2-day time window.

Compare trajectories and FTLE in Areas A and B.

**5. LCS**

Ridge detection

*Python function: LCS.tools.find_ridges_spherical_hessian*

Attracting LCSs

FTLE and attracting LCSs (red ridges)

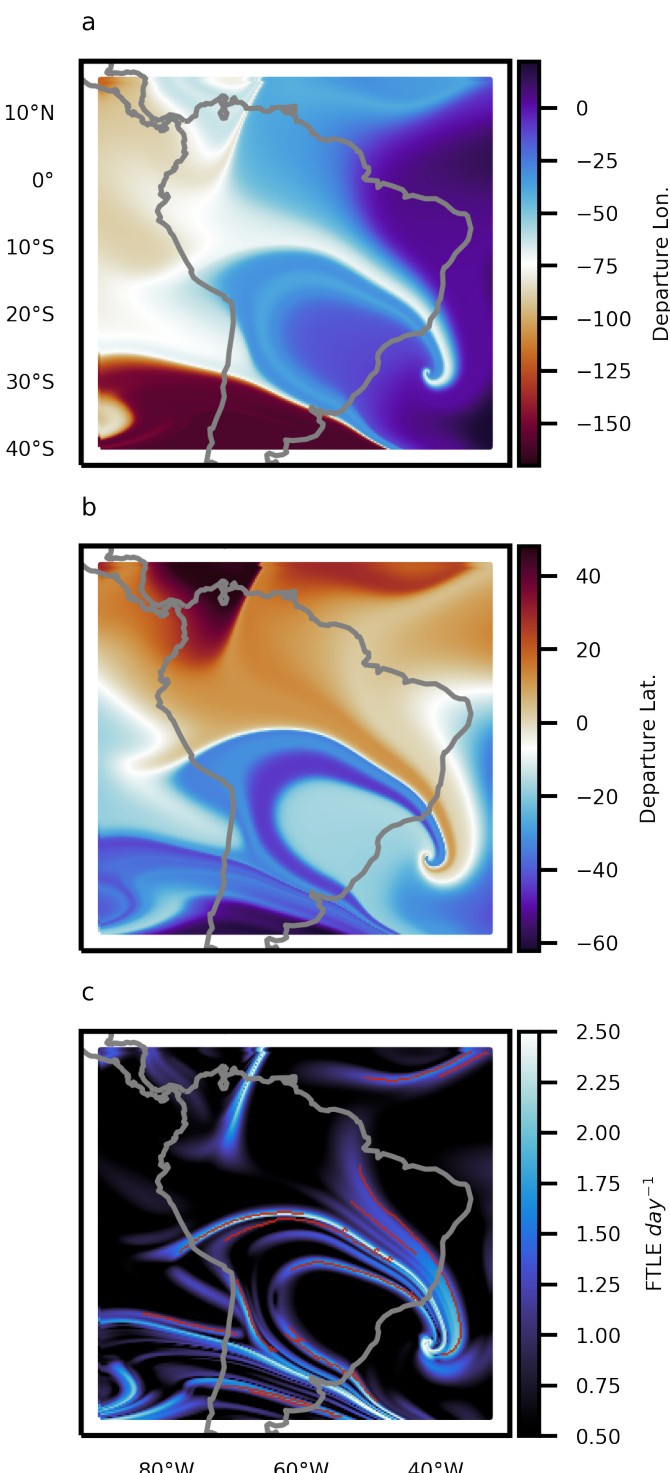

**Figure 4.** Example of the relationship between the meridional (a) and zonal (b) components of the flow-map, the FTLE scalar field and convergence zones (c). Colors in (a) and (b) correspond to the departure longitudes and latitudes, respectively, of trajectories arriving on the ERA5 grid on 24 January 2020 at 18 UTC after advection by $V_{\rho_v}$ for 2 days. Blue shades in (c) correspond to the 2-day FTLE and red lines correspond to height ridges, i.e., attracting LCSs. This day corresponds to the peak activity of an SACZ event.





departure position $\boldsymbol{x_0}(t_0)$ of a parcel to its arrival position $\boldsymbol{x_1}(t_1; \boldsymbol{x_0}, t_0)$ and is given by Equation 2.

$$F_{t_0}^{t_1}(\boldsymbol{x_0}) := \boldsymbol{x_1}(t_1; \boldsymbol{x_0}, t_0) \tag{2}$$

The flow-map can be found at any instant by numerically integrating the trajectories given by:

$$\frac{D\boldsymbol{x}}{Dt} = \boldsymbol{V}_{\rho_v}(\boldsymbol{x}, t) \tag{3}$$

The exponential rate of separation of trajectories departing from the neighborhood of $\boldsymbol{x_0}$ is expressed by the FTLE, represented by $\sigma$:

$$\sigma(\boldsymbol{x_0}) = \frac{1}{|\Delta t|} ln\left(\sqrt{\lambda_{max}(C)}\right) \tag{4}$$

Where $\lambda_{max}(C)$ is most positive eigenvalue of the right-hand Cauchy-Green strain tensor (Eq. 5).

$$C = [\nabla F_{t_0}^t]^T \nabla F_{t_0}^t \tag{5}$$

The gradient of the flow-map $\nabla F_{t_0}^t$ was obtained by a centered finite-difference scheme (Haller, 2001) on spherical coordinates. However, higher accuracy discretizations can be obtained by solving Equation 3 on unstructured meshes with adaptive resolution (Lekien and Ross, 2010).

The norm $\lambda_{max}(C)$ in Eq. 4 essentially quantifies the stretching or folding along the main axis of deformation experienced by a parcel undergoing transformation by the flow-map. If the time trajectories are propagated backwards (i.e., back-trajectories), the FTLE represents the exponential rate of folding; its ridges characterize attracting LCSs. In the forward-in-time case, the FTLE represents the stretching rate and its ridges identify repelling LCSs. As we are interested in the Lagrangian skeletons that potentially organise moisture accumulation and rainfall along a preferred axis, we employ the backwards-in-time convention to identify attracting LCSs as FTLE ridges.

Here we employ an integration time interval of $\Delta t = 2$ days, such that the trajectories are allowed to explore large-scale flow structures. This time scale is also not too long such that the effect of the typical extratropical cyclone, whose average life cycle is three days (Mendes et al., 2010), is filtered out. Although not shown here, features associated with the ITCZ and the SACZ were also found with other integration times (1, 3 and 4 days).

## 2.3 Convergence zones as FTLE ridges

While the scalar FTLE field at a given time characterizes chaotic mixing spatially, its ridges are associated with the most locally intense attraction of back-trajectories arriving in its neighborhood (Haller, 2001; Allshouse and Peacock, 2015). Here we simply define attracting LCSs as curvature ridges of the FTLE scalar field (Shadden et al., 2005). We also impose additional

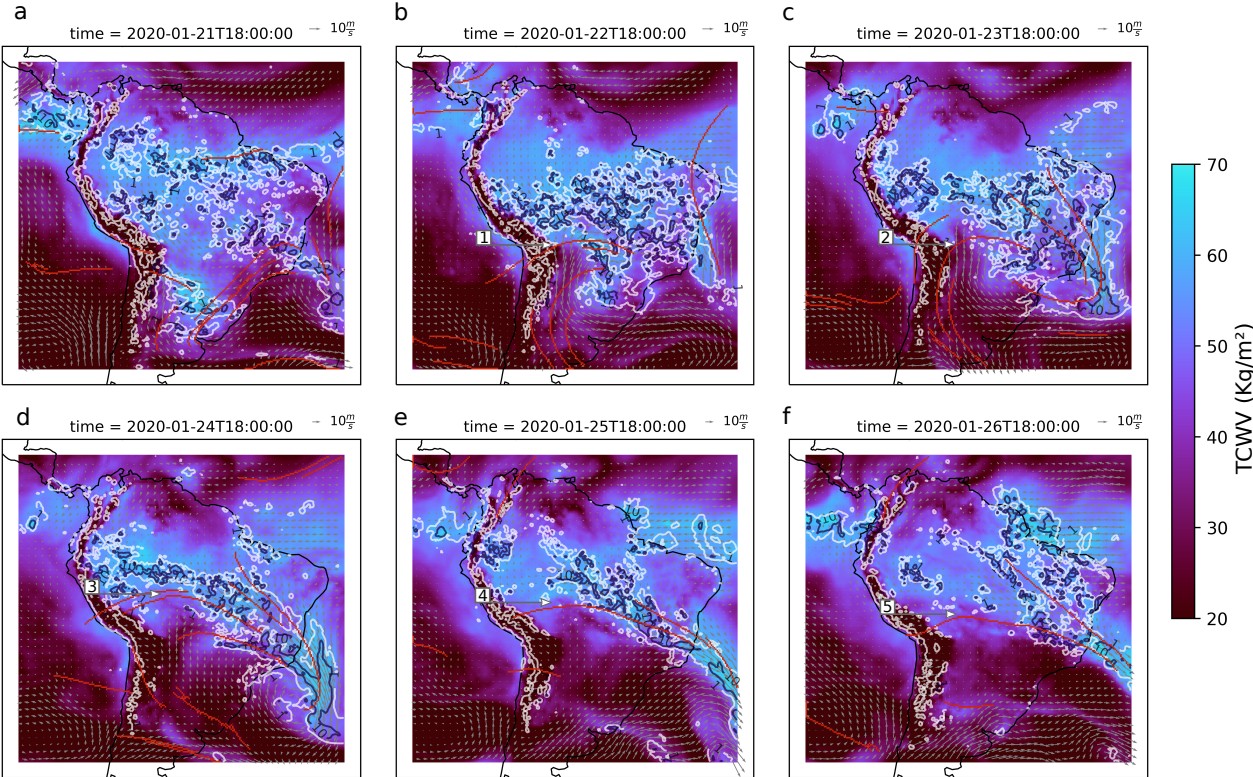

**Figure 5.** Rainfall (contours of 1 and 10 mm/h), vectors of $V_{\rho_v}$ and attracting LCSs (red lines). Time labels are in UTC.

criteria of size and intensity (see Section 3) to further isolate strong attracting structures that are more likely to organise large-scale cloud bands.

Shadden et al. (2005) argue that the flux across curvature FTLE ridges is negligible. This would mean that LCSs defined as such could be regarded as transport barriers. While this is not always true and counterexamples exist (Haller, 2011), FTLE can approximate transport barriers in geophysical flows such as stratospheric circulations (Boffetta et al., 2001), oceanic currents in the Gulf of Mexico (Olascoaga et al., 2006) and subtropical and polar jets (Beron-Vera et al., 2012). Moisture flux anomalies in Section 5 indicate that FTLE ridges in Southeast Brazil act as barriers to important preferential pathways of moisture along
the Andes, leading to negative rainfall anomalies in South Brazil and surroundings.

Therefore, by approximating transport barriers, our definition of convergence zone as attracting LCSs is also consistent with the association of convergence zones to air mass interfaces (Simpson, 1947). Such interfaces can be characterized as sharp gradients of the flow-map that are directly represented in the scalar FTLE field.





## 3 Data and implementation

We compute $\boldsymbol{V}_{\rho v}$ using total column water vapour and the zonal and meridional components of the VIMF at full spatial resolution ($\approx$ 30km) and 6-hourly time intervals from the ECMWF's ERA5 reanalysis between 1980 and 2009. Gridded rainfall data are also obtained from ERA5 at the same spatial and temporal resolutions.

The back-trajectories are calculated by numerically solving Eq. 3 with a two-time-level Lagrangian advection scheme (SET-TLS, Hortal 2002), extrapolated in time iteratively using a second-order Taylor expansion. The velocity $\boldsymbol{V}_{\rho v}$ along the tra-
165 jectories was interpolated with a bivariate spherical spline (Dierckx, 1995). The intensity of the FTLE ridges is dependent on the accuracy of the advection scheme because numerical diffusion can weaken the gradients of the flow-map. The trajectory integration domain was chosen as a wide area ($180^o W/30^o E$, $85^o S/60^o N$) around South America in order to avoid boundary contamination in the domain of interest.

We identified candidate convergence zones as ridges of the FTLE scalar field by relaxing the criteria proposed by Shadden
et al. (2005). The criteria involves isolating curves parallel to the FTLE gradient (condition SR1) and normal to the direction of most negative curvature of the FTLE scalar field (condition SR2). The latter is given by the eigendecomposition of the Hessian matrix. However, in practice, Shadden's criterion is too restrictive (Peikert et al., 2013). Peikert and Sadlo (2008) suggests relaxing SR1 by admitting a tolerance angle ($\epsilon_\theta$) between the curve representing the ridge and the gradient. We employed $\epsilon_\theta = 15^o$. The derivatives for the gradient and the Hessian matrix were computed with a centered-difference scheme on the
sphere.

Convergence zones were obtaining by subsetting the candidate FTLE ridges by size and average intensity. Ridges with average FTLE below $1.2 \, \text{day}^{-1}$ and length shorter than $500$ km were discarded. While these thresholds are arbitrarily defined to filter out weaker and shorter structures, the relative distribution of convergence zones is robust to slight perturbations of the order of $\pm 20\%$. The methodology to compute the FTLE and identify attracting LCSs is summarized in the scheme shown in
Figure 3.

## 4 Interpreting the FTLE scalar field and LCSs

An important distinction in this study is the one between the FTLE scalar field and the attracting LCSs represented by ridges in this field. The FTLE field at a given time depicts the state of mixing in a particular integration time interval. Relatively high FTLE in the backwards-in-time perspective reveals regions where mixing is stronger. In such regions, the departure distance
among parcels back-advected from neighboring arrival points is large. The FTLE represents this exponential folding rate along the principal axis of deformation. Regions where the FTLE is relatively low experience less mixing; i.e., arriving parcels departed from nearby locations. Ridges in this field correspond to the locally strongest attracting structures (i.e., attracting LCSs) associated with strong flow-map deformation. This relationship between trajectory deformation and the FTLE is exemplified in Figure 3.

Figures 4a and 4b represent the zonal and meridional components of the 2-day backwards flow-map on 24 January 2020 at 18UTC. This event was classified by Brazilian meteorology agencies as a particularly strong SACZ associated with intense

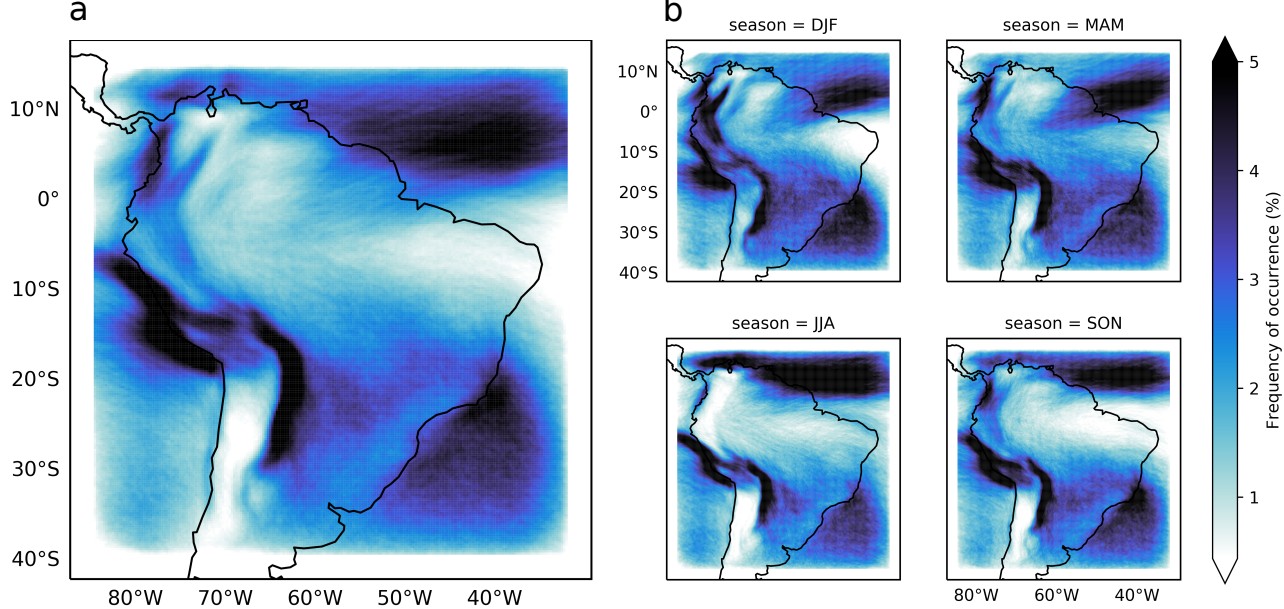

**Figure 6.** Annual (a) and seasonal (b) average frequency of occurrence of convergence zones defined as attracting LCSs between 1980 and 2009.

rainfall in Southeast Brazil (CPTEC, 2020). The gradients of the departure positions represented in Fig. 4a and Fig. 4b are the components of the strain-tensor $C$ used to compute the FTLE scalar field in Fig. 4c. Filaments of high FTLE are immediately evident by visual inspection; ridges corresponding to attracting LCSs are highlighted in red.

The inspection of attracting LCSs and the flow-map components in Figure 4 indicate that convergence zones represent interfaces of initially separated air masses. These interfaces can be visualized as sharp gradients of departure latitudes and longitudes. We highlight the sharp diagonal gradient in Fig. 4b across Brazil: parcels that originated in equatorial latitudes seem to face a transport barrier at about $10^{o}S$; they are deflected east instead of proceeding to southern parts of the continent as they would in the climatological SAMS flow (Fig. 2). This suggests that such structures control the exchanges of moisture
between the Amazon and Southeast/South South America.

## 5 LCSs, moisture and rainfall in a recent SACZ event

Attracting LCSs are structures that shape the evolution of passive tracers in turbulent flows. Atmospheric moisture, however, is not a passive tracer, nor it is homogeneously distributed in the initial times of integration. Therefore, it is not guaranteed that flow entities such as attracting LCSs will shape moisture or rainfall in any meaningful way. In other words, the horizontal
distribution of atmospheric moisture could be simply dominated by local sources and sinks. We know, however, from a climatological perspective, that advection is important for the global hydrological cycle (Trenberth, 1999; Demory et al., 2014). In this section we explore the interplay of attracting LCSs, moisture and rainfall in a recent and significant SACZ event.

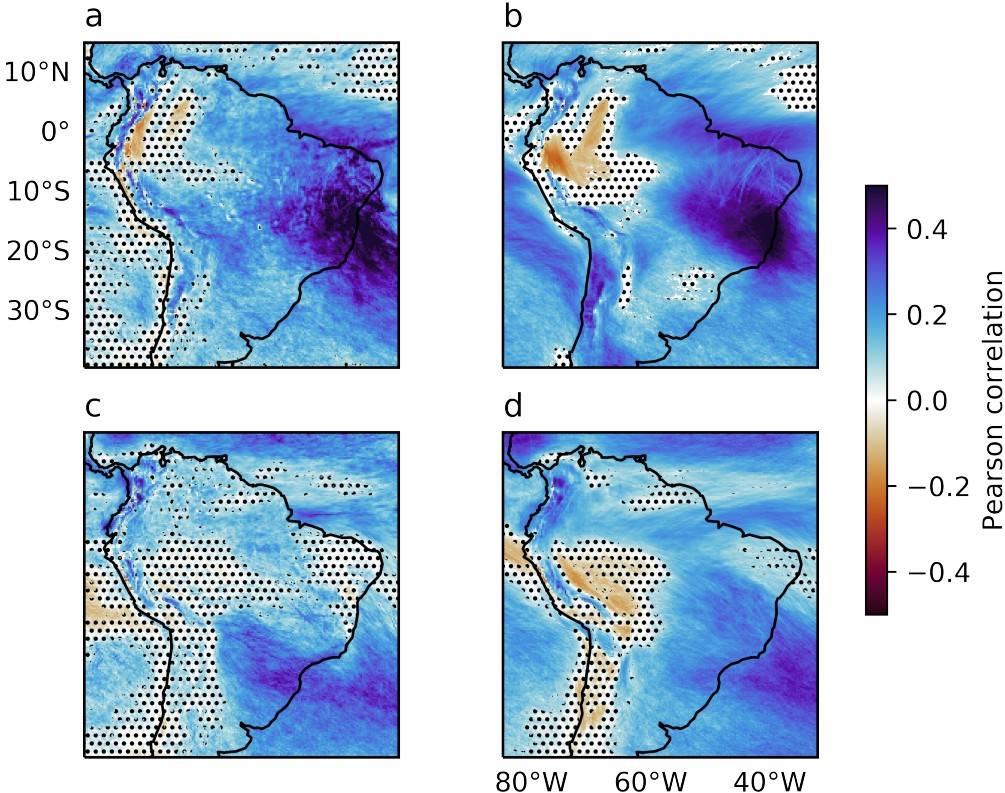

**Figure 7.** Pearson's correlation coefficient between the FTLE and: rainfall (a, c) and TCWV (b, d) in DJF (a, b) and JJA (c, d). Correlations were computed in 6-hourly data. Stippled regions are inside the 99% confidence interval for the null-hypothesis of zero correlation.

Figure 5 shows the evolution of attracting LCSs, TCWV, $V_{\rho_v}$ and rainfall during an SACZ event in January 2020. In Fig. 5a, the SACZ is not yet formed and water vapour is roughly uniformly distributed around the continent. In the equatorial

Atlantic, rainfall and moisture are closely aligned along an attracting LCS revealing the Atlantic ITCZ. In the South Atlantic (bottom-right corner of Fig. 5a), LCSs appear near rainfall a cyclonic circulation. During the next day (5b), an attracting LCS appears in western/central Brazil (Label 1) as an interface along which northward flux from southern South America meets southward flux from the Amazon; they are both deflected east. This interface, identified by the LCS, appears to behave as a transport barrier to the climatological southward Amazonian moisture flux (Figure 2) that supplies moisture to southern South

America. In fact, this barrier (Labels 1 to 5 in Figs. 5b to 5f) persists as TCWV remains low in southern South America.

However, the LCSs in this case study are perhaps most evidently associated with the organisation of moisture and rainfall along a well defined convergence zone. By definition, these LCSs are the locally strongest attracting structures; therefore, we expect that they are the cause of the well-organised rainfall bands in Figures 5d, e and f. The initial situation of uniformly distributed moisture in Fig. 5a and scattered rainfall evolves into a situation where moisture and rainfall are narrowly distributed

along attracting LCSs (Figs. 5d, e and f), forming a single diagonal band across the continent. The whole entity, which we may




call the SACZ, stems from a favorable configuration of the large-scale mixing depicted by a coherent ensemble of attracting LCSs. The attracting LCSs, in their turn, arise from large-scale stirring, caused by the motion of a cyclone.

## 6   Climatology and impact on rainfall and moisture

### 6.1   Frequency of occurrence

The annual and seasonal frequencies of occurrence of convergence zones as defined in Section 2 are shown in Figure 6. A local maximum in the tropical Atlantic coincides with the climatological position of the ITCZ; its frequency increases in austral winter consistent with the increase of rainfall in the ITCZ (de Souza Custodio et al., 2017). Between $10^oS$ and $30^oS$ at $60^oW$, there is a band of higher frequencies along the eastern side of the Andes in all seasons, coinciding with the SALLJ position (Montini et al., 2019). In austral summer, the frequency of convergence zones increases in Southeast Brazil and South Atlantic

near the climatological SACZ position. The local frequency maximum in South Atlantic (approx. $20^oS$ - $40^oW$) is somewhat oriented along the coast. This could result from the interactions of the large-scale flow, sea-breezes and coastal mountain ranges (Fig. 1), which contribute substantially to the wind and rainfall regimes in that region (Silva Dias et al., 1995; Perez and Silva Dias, 2017).

### 6.2   Correlation between the FTLE and rainfall

In this section we investigate how the FTLE correlates with rainfall and water vapour at grid-point scale during austral summer and winter. Since these are Eulerian quantities, a degree of care must be taken when interpreting these correlations. From the moisture budget, changes in the moisture content and rainfall are associated with moisture advection, mass convergence and evaporation in the immediate neighborhood of the atmospheric column considered. The FTLE is a Lagrangian quantity representing the average deformation of arriving trajectories. It is only directly associated with Eulerian mass convergence

in slowly varying or steady flows, where the Lagrangian strain can be approximated by the Eulerian strain (Ottino, 1989). Nevertheless, correlating the FTLE with rainfall and moisture is a simple way to quantify their dependence on mixing.

Figure 7 shows the Pearson's correlation coefficient between the FTLE, rainfall and TCWV at 2-day intervals in summer and winter. The $99\%$ confidence interval was calculated for the null hypothesis of zero correlation. Significant correlations are seen throughout most of the domain, indicating that the moisture content and rainfall in most regions are related, to some degree,

to the level of chaotic mixing. Two features of positive correlations are prominent in summer in Figs. 7a and 7b: a maximum in southeastern/central Brazil and a zonal band in equatorial northern/northeastern Brazil. Rainfall in these regions is directly influenced by the ITCZ and SACZ (Uvo et al., 1998; Ambrizzi and Ferraz, 2015). Negative correlations with TCWV in summer can be seen in western Amazon (approx. $5^oS$, $70^oW$), indicating that the rainfall and moisture content rely on local sources or non-mixed (parallel trajectories) transport. In winter (Figs. 7c and 7d), the FTLE is positively correlated with rainfall in South

Brazil; Paraguay and Uruguay are the most correlated with rainfall, while the storm-track in the South Atlantic is where the positive correlation of FTLE with TCWV is highest.



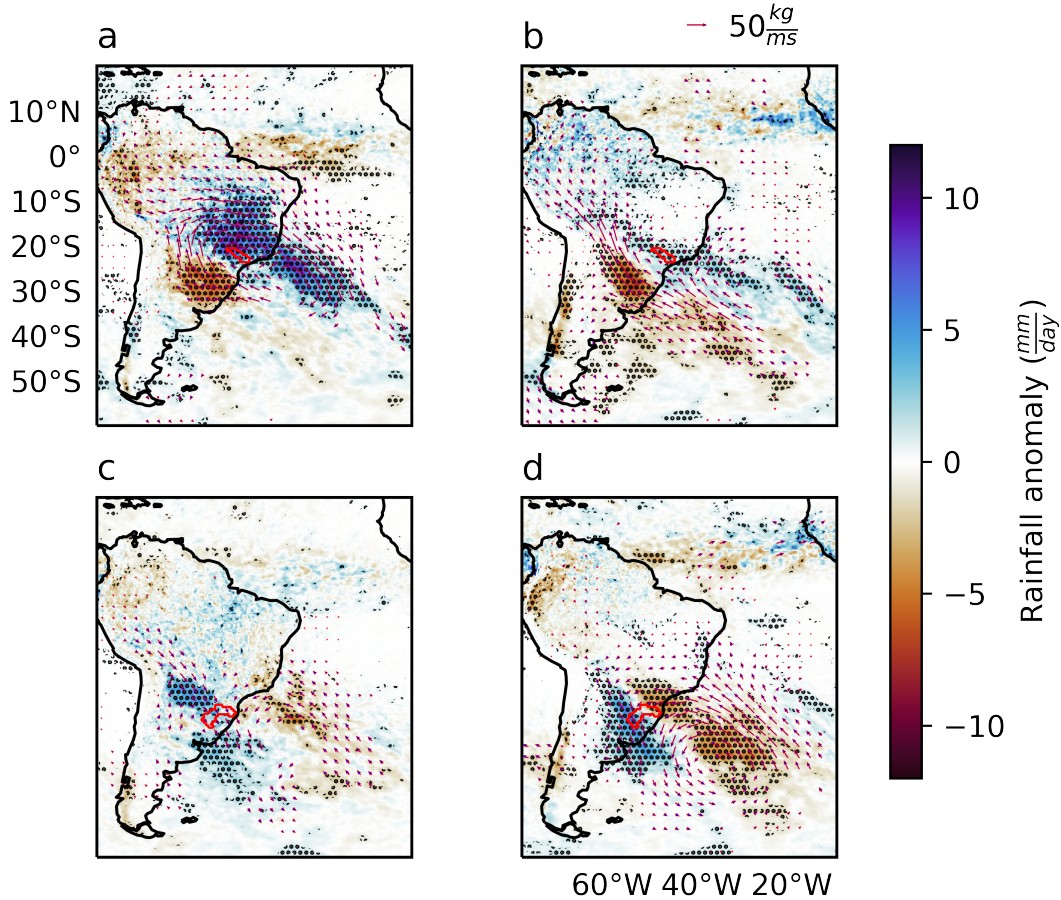

**Figure 8.** Anomalies of rainfall and VIMF (vectors) during attracting LCS events in the Tietê (a, b) and Uruguay (c, d) basins in DJF (a, c) and JJA (b, d). Anomalies significant at the 99% confidence level are stippled.

### 6.3 Moisture flux and rainfall anomalies associated with attracting LCSs

To investigate how the location of attracting LCSs influences rainfall and moisture transport variability, we focus on two watersheds that are tributaries of the La Plata River: Tietê and Uruguay. These basins were chosen because they are typically

influenced by two of the mechanisms of interest: the SACZ in Tietê and the SALLJ in Uruguay. Economic activity in both watersheds depend on rainfall particularly due to hydroelectric power generation and agriculture. Moreover, a number of densely populated cities, including São Paulo, are located in the Tietê catchment area.

Figure 8 shows the anomalies of rainfall and VIMF in summer and winter during events of attracting LCSs over the Tietê and Uruguay watersheds. The events were selected when LCSs intersected the watershed areas. In summer in the Tietê basin (Fig.

8a), attracting LCS events are associated with a significant anomalous rainfall dipole and cyclonic VIMF. North of the basin,



**Figure 9.** Geopotential height anomalies at 250 hPa during attracting LCS events in the Tietê (a, b, c and d) and Uruguay basins (e, f, g h) during the austral summer (DJF) and winter (JJA) for lags 0 and -3 days. Anomalies significant at the 99% confidence interval are stippled. The watershed boundaries are in red contour.

positive rainfall anomalies form a diagonal band extending from central Brazil to the South Atlantic; to the south, there is a negative rainfall anomaly. Similar anomalies were associated with the SACZ by Muza et al. (2009). A northward anomalous flux along the Andes indicates a weaker SALLJ during attracting LCS events in Tietê. This is also noticed in winter (Fig. 8b),





although the positive rainfall anomalies are weaker and the cyclonic anomaly is centered in the South Atlantic. The SALLJ
weakening could produce the significant negative rainfall anomalies near South Brazil and Paraguay. This interplay between
the SACZ and SALLJ is a documented feature of the South American climate (Boers et al., 2014).

During LCS events in the Uruguay basin (Figs. 8c and 8d), there is an anti-cyclonic VIMF anomaly in the South Atlantic
associated with a southward flux along the Andes, indicating stronger SALLJ. The rainfall anomalies are positive near South
Brazil, Uruguay and Paraguay. Liebmann et al. (2004) found similar rainfall anomalies of about 4 mm/day in the vicinity of
the Uruguay basin during strong SALLJ events.

The consistency of the anomalies in Figure 8 with previous studies about the SACZ and SALLJ highlights the potential of
the methodology to identify important mechanisms of moisture transport. Moreover, it indicates that the SACZ and the SALLJ,
from a kinematical point of view, are similar coherent structures stemming from large-scale turbulence. Once the structures are
identified under the same framework, we may ask which anomalies they cause and what dynamics originate and sustains these
distinct patterns of mixing. This differentiates our approach from studies that identify these structures individually based on
existing expectations of their impact on rainfall and cloudiness or previous understanding of the forces involved.

### 6.4    Geopotential anomalies

Figure 9 shows the geopotential anomalies at 250 hPa in summer and winter during LCS events in the Tietê and Uruguay
basins. During events in Tietê in summer (Figs. 9a and 9c), an anomalous trough is positioned south of the watershed roughly
aligned with the cyclonic circulation in Figure 8a. This trough is connected with a wave pattern that appears to propagate from
the South Pacific (Fig. 9c). This observation is consistent with the mechanism of SACZ formation proposed by Van Der Wiel
et al. (2015) based on ray-tracing diagnostics of Rossby waves initiated by a heat source in the same location.

In winter in Tietê (Fig. 9b and 9d), a wave pattern arises similar to the Pacific-South Atlantic teleconnection (Ambrizzi et al.,
1995), positioning a trough south of the Tietê watershed. This trough is slightly to the east (downstream) of the anomalous low-
level circulation in Figure 8b, indicating baroclinicity. This type of configuration has been noted as a cyclogenesis mechanism
in Southeast and South Brazil (Crespo et al., 2020).

During LCS events in the Uruguay basin in both seasons (Figs. 9e and 9f), an anomalous 250 hPa ridge is positioned in the
South Atlantic, aligned with the anti-cyclonic circulation in Figures 8c and d. The lagged anomalies (Figs. 8g and 8h) indicate
that this ridge originates from a perturbation in polar latitudes.

## 7    Summary and conclusions

This is the first study to investigate the role of Lagrangian Coherent Structures (LCSs) in tropical and subtropical rainfall. We
defined skeletons of atmospheric convergence zones as attracting LCSs given by ridges of the Finite-time Lyapunov Exponent
(FTLE). The FTLE is a measure of deformation among neighboring trajectories that synthesizes the state of mixing in fixed
time intervals and allows the visualization of transport barriers. Defining convergence zones as attracting LCSs is consistent
with a Lagrangian understanding of convergence zones as regions where remotely sourced air masses interact (Simpson, 1947).



This definition also implies that convergence zone skeletons are associated with tracer accumulation, potentially explaining the organization of cloud and rainfall bands.

Attracting LCSs frequent tropical and subtropical South America, with climatologies consistent with previous studies of large-scale phenomena such as the Intertropical Convergence Zone, the South Atlantic Convergence Zone (SACZ) and the South American Low-Level Jet (SALLJ). Point by point correlations showed that, in the typical areas of action of these mechanisms, moisture and rainfall depend to some extent on the large-scale mixing represented by the FTLE scalar field.

Fixing locations of interest in watersheds in South and Southeast Brazil, we showed that significant rainfall and moisture flux anomalies are associated with attracting LCS events. These anomalies are consistent with previous climatologies of the SACZ and the SALLJ. We also analysed geopotential anomalies at 250 hPa during LCS events in these two basins. The geopotential composites suggest that remotely sourced perturbations are stirring the low-level synoptic-scale flow such that attracting LCSs arise and organise the moisture transport and rainfall in fine bands. This behaviour resembles the spectrally non-local regime in the stratosphere, where the evolution of tracer features is set by low wavenumber perturbations (Shepherd et al., 2000).

In the case of the SACZ, our approach differs substantially from existing classifications that equate the SACZ to patterns associated with convective or rainfall variability (Carvalho et al., 2004; Ambrizzi and Ferraz, 2015). Our approach first asserts the existence of coherent flow features organising atmospheric moisture. Only then do we associate them with rainfall variability. However, our results are compatible to a good degree with rainfall and circulation anomalies found in previous studies, suggesting that the methodology can be employed as a criterion for SACZ events.

For future developments, we suggest refining the classification by allowing different types of convergence zones. This could be done by analysing physical quantities across and along the attracting LCSs. For example, if the temperature gradient is strong perpendicular to the FTLE ridge, such convergence zone could be associated with a frontal system. Similarly, a convergence zone could be associated with orography if the LCS is parallel to elevation features. Alternatively, convergence zones could be filtered by properties of the arriving parcels, such as temperature and water vapour concentration, likewise the trajectory filtering available in LAGRANTO (Sprenger and Wernli, 2015). This refined classification would help understanding the mechanisms that generate the LCSs and, ultimately, organised rainfall bands. As a final goal, the proposed framework could serve as basis for a global criterion for convergence zones, replacing or being combined with region-specific classification methods.

*Code and data availability.* All datasets used in this study are publicly available at the ECMWF web platforms. The Python libraries for trajectory computation and LCS identification are also publicly available at *https://github.com/gabrielmpp/LagrangianCoherence*

*Author contributions.* Author Gabriel Perez is the main contributor in the conceptualization, implementation, analysis and writing of this research paper. Authors Pier Luigi Vidale and Nicholas Klingaman supervised the development of this research as well as contributed with





conceptualizing and writing the manuscript. Author Thomas Martin contributed with software development and quality control as well as data visualization.

*Competing interests.* The authors declare that they have no conflict of interest.

*Acknowledgements.* This study was financed in part by the Coordenação de Aperfeiçoamento de Pessoal de Nível Superior – Brazil (CAPES)
– Finance Code 88881.170642/2018-01. Authors Pier Luigi Vidale and Nicholas Klingaman acknowledge the National Centre for Atmospheric Science (NCAS, UK). Author Thomas Martins acknowledges the Department of Atmospheric Sciences of the University of São Paulo (IAG-USP, Brazil).




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
