# Peer review of "Atmospheric convergence zones stemming from large-scale mixing"

_Weather and Climate Dynamics, 2020_

## Referee Comment (RC1) · Anonymous Referee #1 · 24 Feb 2021

**1 Introduction**

The authors analyze the convergence zones and precipitation patterns in South America, using an approach coming from the dynamical systems and Lagrangian analysis: the Lagrangian Coherent Structures. This approach detects the stable/unstable manifold in a time-dependent flow trying to find the skeleton or the main features in the fluid flow that organize the transport. The authors employed here the LCS detection method based on the FTLE ridge extraction with some approaches. The authors focus on a case study of a precipitation rainfall event over Brasil and try to characterize this event in terms of LCS after describing the methodology employed. Then analyze further from a climatology point of view the link between the persistence of this attractive

LCS with rainfall patterns. To that end, they perform a correlation between the presence of LCS and rainfall and other variables. Finally, to analyze the synoptic meteorological mechanism that shapes the wind flow and hence the convergence, they analyze the role that the geopotential height anomalies play. The authors found that exists a link between the LCS and the rainfall patterns on that area.

General comments I found the article interesting and must be considered for publication, however, I found key points that should be addressed before publication. The main points are that, in my opinion, there is a lack of connection between sections, I do not find a true motivation to go from a case study to a climatological study in the introduction. At this moment, it looks more like a sequence of isolated sections more than a logical connection between them. I suggest the authors introduce a short paragraph in the introduction, briefly explaining the motivation to go from a case study to a climatic study. For example, maybe the authors are interested in analyzing the impact of LCS through the different time-scales or how the mixing patterns affect the local and climatic scales. Also, I do not found a connection between the FTLE or LCS analysis for a case study (Section 5) and the extension to a climate analysis (Section 6). What period do the authors analyze? How do they extend this analysis from a local case? Do they do the same analysis at every time step? Finally, in my opinion, the conclusions should be improved. A simple characterization of this phenomenon in terms of LCS is a poor conclusion. I suggest the authors comment on the potential applications of this methodology to find convergence patterns in other regions and, more importantly, their viability as a forecast indicator.

**2 Specific comments**

**Line 22: "Despite these historical associations with coherent trajectories".** In my opinion, the authors should consider the use of "winds" instead of trajectories. The

"convergence zones" are the consequence of coherent winds.

**Line 23: "…frequently identify by "heuristic rules".** Do the authors consider that the previous authors focused more on analyse the results qualitatively? or, do the previous authors base on arbitrary thresholds based on the experience?

**Line 130: Cauchy-Green Formulation** The spherical formulation of the FTLE based on the Cauchy Green deformation tensor includes and additional transformation matrix. Could the authors check if this formulation included in Haller 2001? As far as I know, this formulation is done in https://doi.org/10.1016/j.physd.2012.06.012.

**Line 169: We identified candidate convergence zones as ridges of the FTLE scalar field by relaxing the criteria proposed by Shadden et al. (2005). The criteria involve isolating curves parallel to the FTLE gradient (condition SR1) and normal to the direction of most negative curvature of the FTLE scalar field (condition SR2). The latter is given by the eigendecomposition of the Hessian matrix. However, in practice, Shadden's criterion is too restrictive (Peikert et al., 2013). Peikert and Sadlo (2008) suggest relaxing SR1 by admitting a tolerance angle () between the curve representing the ridge and the gradient. We employed**$(\epsilon_\theta = 15)°$

The authors are right in their approach however I found it a very strong approach. The authors go from $90°$ to $15°$) without more explanation. For example, how do the authors ensure that the detected LCSs on the FTLE field are not close to shear stress between the Andes mountains and the neighbouring wind flow as we can see in some LCS like in Figure 5, b,c? Did the authors perform a parameter analysis changing those values? Could the authors explain this choice very shortly?

**Line 176: "subsetting the candidate FTLE ridges by size and average intensity. Ridges with average FTLE below 1.2 day-1 and length shorter than 500 km were discarded. While these thresholds are arbitrarily defined to filter out weaker and shorter structures, the relative distribution of convergence zones are "robust to**

**slight perturbations of the order +-20**$\%$

Do the authors refer to the previously mentioned parameters? The authors do a great job presenting a flow map of the work. I suggest the authors should include a short table within the main parameters employed to analyse these phenomena with T, $\epsilon_\theta$, Ridge intensity and length of the LCS.

**Figure 6**: In my opinion, the authors should improve what is the meaning of the "frequency of occurrence" of LCS? Is the number of timesteps that LCS was detected at grid-cell divided by the number of total time steps? $\frac{t_{iLCS}}{t_i}$

**Line 202: "Attracting LCSs are structures that shape the evolution of passive tracers in turbulent flows."**

They do not have necessary to be turbulent flows. Is the double gyre from Shadden a turbulent flow? There is no cascade of energy transfers frow higher flow scales to the lower ones, however, there are attracting LCS. The author should consider using "in turbulent or chaotic flows" or more general "in time-dependent flows".

**Line 210-214. The authors mentioned the LCS on figure 5 such as convergent LCS.** It looks like this LCS is a consequence of the different wind flow of the particles over the Andes where the speed seems to be lower (close to 0) with the speed on the neighbourhood. This causes a shear LCS, between the Andes and the surrounding wind flow. In my opinion, this is the reason why the authors have a 0-Pearson correlation coefficient with rainfall over that area. The LCS are not true LCS. However, the deflected flow goes from south to North converging with the Northwind being a truly attractive LCS. Could the authors clarify this point?

**Line 241: "correlating the FTLE with rainfall and moisture is a simple way to quantify their dependence on mixing."** Correlate the FTLE against rainfall one-one is does not seems an indication that you are linking their dependence against mixing. In my opinion, to talk about mixing, you should use an integrated measure based on

the FTLE because the time variability of the FTLE is what introduces mixing, not instantaneously value. Could the authors clarify the choice of this method or also cite previous work where this methodology has been used?

**3 Technical comments.**

The quality of the figure should be improved in general. The authors should consider adding the labels, and at least the minor ticks on these figures.

**Figure 8**: The authors should indicate what red polygon means. I suggest to the authors append to the left a figure with the composition of the LCS detected for all these events.

**Figure 4**: The y-labels should be placed on each figure or at least the minor grid ticks to know the association between the data. In any case, instead of the use of "departure lat", I suggest appending the mathematical formulation from where the FTLE derives. It is the flow map evaluated at a time r(t) = $\phi_{t_0}^{T-t_0}(r(t_0), t_0)$

---

## Short Comment (SC1) · 5 Mar 2021

I appreciate the authors' interesting take on large-scale mixing in their submission. The submission tackles an impressive range of topics on the SACZ and kinematics, with a wide range of cited literature. My hat is off to the authors for assembling such an impressive range of scientific content.

I have broken my review into three parts.

A. The first part relates to literature that should be cited.

1. Other derivations of quantities that serve the same function or are similar to the FTLE have been published, but have not been described by the authors.

[Figure]

Okubo, A., 1970: Horizontal dispersion of floatable particles in the vicinity of velocity singularities such as convergences. Deep-Sea Res., 17, 445–454

McWilliams, J. C., 1984: The emergence of isolated coherent vortices in turbulent flow. J. Fluid Mech., 146, 21–43.

Benzi, R., S. Patarnello, and P. Santangelo, 1988: Self-similar coherent structures in two-dimensional decaying turbulence. J. Phys. A: Math. Gen., 21, 1221–1237.

Weiss, J., 1991: The dynamics of enstrophy transfer in two-dimensional hydrodynamics. PhysicaD, 48, 273–294

Cohen, R. A., and C. W. Kreitzberg, 1997: Airstream boundaries in numerical weather simulations. Mon. Wea. Rev., 125, 168–183.

Cohen, R. A., & Schultz, D. M. (2005). Contraction Rate and Its Relationship to Frontogenesis, the Lyapunov Exponent, Fluid Trapping, and Airstream Boundaries, Monthly Weather Review, 133(5), 1353-1369.

Arnup, S. J., & Reeder, M. J. (2007). The Diurnal and Seasonal Variation of the Northern Australian Dryline, Monthly Weather Review, 135(8), 2995-3008.

2. Although not directly related to the FTLE, climatologies of upper-level cut-off lows and moisture flows into South America have been discussed recently by Muñoz and collaborators. The second may be more relevant to this manuscript.

Muñoz, C., D. M. Schultz, and G. Vaughan, 2020: A midlatitude climatology and interannual variability of 200- and 500-hPa cut-off lows. J. Climate, 33, 2201–2222, doi: 10.1175/JCLI-D-19-0497.1.

Muñoz, C., and D. M. Schultz, 2021: Cut-off lows, moisture plumes, and their influence on extreme precipitation days in central Chile. J. Appl. Meteor. Clim., https://doi.org/10.1175/JAMC-D-20-0135.1.

3. There is a whole body of atmospheric river literature led by Reginald Newell and

Marty Ralph that is not cited here. Ralph, in particular, is credited with promoting the atmospheric river concept in the past twenty years and publishing dozens of papers on the topic. I am sorry to see that the authors did not cite a single one of his papers, despite being the leading authority on atmospheric rivers. There is also a book by Springer on atmospheric rivers, recently published.

Moreover, a recent study by Valenzuela and Garreaud (2019) presents the linkage between atmospheric rivers and heavy precipitation in Chile.

Valenzuela, R. A., and R. D. Garreaud, 2019: Extreme daily rainfall in central-southern Chile and its relationship with low-level horizontal water vapor fluxes. J. Hydromet., 20 (9), 1829–1850, doi:10.1175/JHM-D-19-0036.1.

See also the following article.

Viale, M., R. Valenzuela, R. D. Garreaud, and F. M. Ralph, 2018: Impacts of atmospheric rivers on precipitation in southern South America. J. Hydrometeor., 19, 1671–1687, doi:10.1175/ JHM-D-18-0006.1.

4. The authors discuss the concept of airmass interfaces at line 156–158, but fail to discuss the previous literature on air masses, airstreams, and airstream boundaries. In addition to the Cohen papers already mentioned, there are the airstream models of Carlson (1980) and Browning (1990), among many others, let alone the concepts of airmass analysis discussed by the Bergen School meteorologists, especially Bergeron (1928, 2020) (Schultz et al. 2020 discusses the importance of this article to airmass analysis and interpretation in more detail).

Carlson, T. N., 1980: Airflow through midlatitude cyclones and the comma cloud pattern. Mon. Wea. Rev., 108, 1498–1509.

Browning, K. A., 1990: Organization of clouds and precipitation in extratropical cyclones. Extratropical Cyclones: The Erik Palme Ìąn Memorial Volume, C. W. Newton and E. O. Holopainen, Eds., Amer. Meteor. Soc., 129–153.

Bergeron, T., 2020: Three-dimensionally combining synoptic analysis. First part: Fundamental introduction to the problem of airmass and front formation [originally published as Über die dreidimensional verknüpfende Wetteranalyse. Erster Teil: Prinzipielle Einführung in das Problem der Luftmassen- und Frontenbildung, Geofys. Publ., V (6), 1930]. Edited by David M. Schultz, translated by Gerald Prater, Bull. Amer. Meteor. Soc., 101 (Suppl.), doi: 10.1175/BAMS-D-20-0021.2.

Schultz, D. M., H. Volkert, B. Antonescu, and H. C. Davies, 2020: Defender and expositor of the Bergen methods of synoptic analysis: Significance, history, and translation of Bergeron's (1928) "Three-dimensionally combining synoptic analysis". Bull. Amer. Meteor. Soc., 101, E2078–E2094, doi: 10.1175/ BAMS-D-20-0021.1.

5. A very similar figure to Fig. 6 was published by Thomas and Schultz (2018, Fig. 15). Your manuscript should discuss this comparison. More generally, the discussion of how to analyze airmass boundaries and airstream boundaries are more thoroughly discussed in this article, which would seem to be highly relevant to your manuscript. Some discussion of these issues, citing this article as needed, should likely be included in your manuscript.

Thomas, C. M., and D. M. Schultz, 2019: Global climatologies of fronts, airmass boundaries, and airstream boundaries: Why the definition of "front" matters. Mon Wea. Rev., 147, 691–717, doi: 10.1175/MWR-D-18-0289.1.

This article may also be relevant.

Thomas, C. M., and D. M. Schultz, 2019: What are the best thermodynamic quantity and function to define a front in gridded model output? Bull. Amer. Meteor. Soc., 100, 873–895, doi: 10.1175/BAMS-D-18-0137.1.

6. There is an abundant and important literature on the SACZ that is not cited. I'm not familiar with all of this literature, but the authors should see if any of the results from these articles (as well as others) are relevant to their own conclusions in sections 5

and 6.

Robertson, A. W., & Mechoso, C. R. (2000). Interannual and Interdecadal Variability of the South Atlantic Convergence Zone, Monthly Weather Review, 128(8), 2947-2957.

Liebmann, B., Kiladis, G. N., Marengo, J., Ambrizzi, T., & Glick, J. D. (1999). Submonthly Convective Variability over South America and the South Atlantic Convergence Zone, Journal of Climate, 12(7), 1877-1891.

Grodsky, S. A., & Carton, J. A. (2003). The Intertropical Convergence Zone in the South Atlantic and the Equatorial Cold Tongue, Journal of Climate, 16(4), 723-733.

Liebmann, B., Kiladis, G. N., Vera, C. S., Saulo, A. C., & Carvalho, L. M. V. (2004). Subseasonal Variations of Rainfall in South America in the Vicinity of the Low-Level Jet East of the Andes and Comparison to Those in the South Atlantic Convergence Zone, Journal of Climate, 17(19), 3829-3842.

Carvalho, L. M. V., Jones, C., & Liebmann, B. (2002). Extreme Precipitation Events in Southeastern South America and Large-Scale Convective Patterns in the South Atlantic Convergence Zone, Journal of Climate, 15(17), 2377-2394.

Castro Cunningham, C.A. and De Albuquerque Cavalcanti, I.F. (2006), Intraseasonal modes of variability affecting the South Atlantic Convergence Zone. Int. J. Climatol., 26: 1165-1180. https://doi.org/10.1002/joc.1309

B. The second part deals with the consistency of this manuscript with the previous literature.

1. Is VIMF the same as IVT, used in the atmospheric river literature? If so, stick with conventional terminology already published. If a quantity has already been published, please don't introduce new terminology to describe the same features. Don't make the literature more impenetrable than it already is. If not, relate your quantity to previously published metrics of moisture transport.

2. In the same vein is the introduction of the quantities such as the strain tensor. It seems these topics have been published before by others. Follow consistent variable names with the previous literature for ease of comparison, where possible. If your derivation comes from previous literature, please cite that literature more closely.

3. Cohen and Kreitzberg (1997, section 4d) used 12 h in their calculations to understand airstream boundaries, but you chose 2 days. They explained their rationale in a more substantive way than you did. Perhaps some reference to their discussion is worthwhile, as well as a comparison between the differences in your respective choices.

C. The third part presents other concerns related to terminology and readability.

1. The title gives the appearance of a much more general study than the one that is presented. The abstract clarifies that the authors specifically are interested in the "role of large-scale turbulence in shaping atmospheric moisture in South America." It would seem to make sense that "moisture" and "South America" appear in the title. Doing so will ensure that the authors find the most interested and appropriate readers for their manuscript. That's a win–win for readers and authors!

2. Through the mass continuity equation, low-level convergence zones require ascent. Thus, the implication that "organized cloud bands...are often regarded as convergence zones" and that "flow kinematics are not usually taken into account" (lines 2–4) seems a bit unusual. Convergence is a kinematic quantity, so I'm unclear on the point here. Furthermore, it would make sense for the authors to downplay this association because of the clear relationship between convergence and ascent (and hence, clouds and precipitation). Deformation cannot produce ascent through direct kinematic effects. There are places in the manuscript where I get the sense the authors understand this, but this message could be presented more clearly, helping to make the point that I believe that they are trying to make.

3. I found the large number of acronyms difficult to follow as the paper went on.

Some terms do not require introduction of an acronym. Too many unfamiliar acronyms makes it difficult for readers to follow your manuscript, as readers encountering an acronym that they can't remember are left to flip back through the paper to track it down. Also, acronyms make it difficult for readers who don't read the manuscript linearly from introduction to conclusion, but instead skip around through the manuscript to get the relevant information they require. Please eliminate many, if not all, nonstandard acronyms when you revise your manuscript. Doing so will improve the readability of your manuscript.

4. I had a hard time navigating the introduction. It seemed to be a series of paragraphs on various topics and citations rather than a coherent narrative that motivated the paper and got me as a reader interested in pursuing it further. Consider section 1.2, just as an example. The paragraphs appear to be about the following topics.

1. general South American geography and atmospheric rivers

2. Amazon convection and SAMS

3. extratropical cyclones and cold fronts

4. cloudiness algorithms and problems with EOFs

There is no flow or coherence between these paragraphs. I don't see how these topics come together to form a unified subsection on "Aspects of the moisture transport in South America". This unity and coherence needs to be improved and the organization of the introduction needs to be rethought. Please read Gopen and Swan (1990) for improving your coherence.

https://www.americanscientist.org/blog/the-long-view/the-science-of-scientific-writing

5. The purpose of this paper is only stated in the last sentence of the introduction, 100 lines in. "This study is motivated by the need to objectively link convergence zones to atmospheric flow features, which cannot be done using existing definitions." Furthermore, this statement hasn't been justified sufficiently for the readers to have buy-in

at this point in the manuscript. Simply put, this statement has not been sufficiently motivated at this point in the manuscript, further evidence that the introduction needs reorganization and rethinking.

6. I would caution the authors about using "objective" when they really "automated". In reality, most objective techniques are quite subjective. See section 18.2 in Schultz (2009) for further discussion of this distinction.

Schultz, D. M., 2009: Eloquent Science: A Practical Guide to Becoming a Better Writer, Speaker, and Atmospheric Scientist. American Meteorological Society, 412 pp. http://bit.ly/EloqSci.

7. Lines 156–157: I didn't really get the sense that this paragraph was a well-reasoned discussion of this issues. More development of these concepts is needed.

8. Section 7 is called "Summary and conclusions". What's the difference between summary and conclusions? Could just one word suffice?

9. I found section 7 a bit disappointing. It needs a better organization and narrative to tell the story of what the authors found.

10. In that regard, maybe it is the structure of the rest of the paper, but it seemed more like a series of sections that were not very well connected. I'm not sure what the solution is, but any revisions to the manuscript that the authors could make to tell a better, more engaging story would help. I feel that they know the narrative, but might they be assuming that the readers know as much as they do and automatically know the issues that they face and wish to address? In some ways, that's how it feels to me. Just remember that not all readers are intimately familiar with the issues that you are trying to solve. Take a step back in your explanations and present it to us in a way that makes us understand the same difficulties with the state of the science that you already know.

---

## Referee Comment (RC2) · Anonymous Referee #2 · 13 Mar 2021

**Review of**
**Atmospheric convergence zones stemming from large-scale mixing**

Gabriel M. P. Perez, Pier Luigi Vidale, Nicholas P. Klingaman, and Thomas C. M. Martin

March 18, 2021

**General**

The authors identify and investigate convergence zones in terms of Lagrangian Coherent Structures. Mathematically, Fine-Time Lyapunov Exponent is used for the classification. By evaluating climatologies the results of the authors are consistent with previous investigations of the Intertropical convergence zone, the South Atlantik Convergence Zone and the South American Low-Level Jet. Furthermore, on smaller, regional scale, the authors show that rainfall and moisture flux anomalies are associated with LCS events.

The novel approach of the authors for the study of Convergence zone is interesting and motivate for further applications as it is suggested in the end of the conclusions.

Your approach might also be interesting for interdisciplinary working scientists from Meteorology, Mathematics, Informatics or Physics, who are all in the community investigating algorithms and applications for coherent structures. Therefore, it would be helpful to explain, where the meteorological abbreviations stand for, e.g. TCWV (figure 5 and text) is not defined. Please make sure to define every abbreviation you use. Moreover, I recommend to include in the title the words "coherent structure" to have a wider readership. I appreciate the citations of the very classical works!

I recommend to publish this work in Weather Climate Dynamics after correcting some minor issues listed below.

**Minor comments:**

- p.2 line 45ff. Could you explain a litte bit more in detail how do you define coherence/coherent structures in terms of FTLE

- Further approaches do not count elongated structures as coherent sets, why does this work here?

- Are there further approaches to identify coherent structures of meteorological phenomena besides FTLE?

- fig 4: are the departure positions or the gradients of the departure positions are shown? Which unit does they have? Are they components of x0? The figure should be placed near the description in the text.

- TCWF is not defined (e.g. fig 5)

- figures and explanation of the figures are often placed far away in the paper, please check. E.g. fig. 5 on page 9, but explained on page 12, fig 4 is explained much later

- figure 5: How are the LCS calculated? Nice example, please explain more in detail on this example referring to the used formulas.

- Page 5: Would suggest to not end the subsection with a formula and place the formula before the sentence 'Physically, V is the average flow by which the total column (...)'

- Page 8: Since $V_{\rho_v}$ is vertically integrated, I guess $V_{\rho_v} = V_{\rho_v}(x, y)$ such that $\mathbf{x_0}$ and $\mathbf{x_1}$ are in $\mathbb{R}^2$ ?

- Page 8: you integrate trajectories (2)-(3), but how do you identify the trajectory?

- Page 8, line 126 ff: why is the rate sigma exponential and not logarithmic (the sentence before that formula let us assume that $\sigma$ is the exponential rate)?

- Page 8: why is $\lambda_{max(C)}$ at the same time an Eigen value (line 129 p.8)and a norm (line 134 p.8)?

- page 8: $\Delta t = 2$ days as time resolution. Did you use this time resolution for all results showing in the paper?

- page 10, l.160 please refer to (1)

- how does the method handle sources and sinks?

- page 13: you show the frequency of occurrence in $\%$, what is the absolute number of events (approximately)?

- p.13. beginning of 6.2 could you add 1 sentence what is the motivation of this comparison?

- figure 8 larger? The VIMF is hardly recognizable, but very interesting results.

- p.17 l. 303-304 (. . .) are consistent with pervious climatologies (. . .) please add sources

- p.17 l. 311-312 please add sources

---

## Author Comment (AC3) · 27 Mar 2021

article

**Reply to David Schultz**

Gabriel M P Perez

March 2021

We appreciate the interest of Dr. Schultz in our research article and thank for his insightful comments. We will address the comments we judge more relevant for the WCD's audience below.

Dr. Schultz suggests that we use the word "automated" instead of "objective" to describe our methodology. We strongly disagree with this. We suggest the reading of the seminal work of Shadden et al. (2008) on Lagrangian Coherent Structures (LCSs). The authors demonstrate the objectiveness of FTLE ridges by deriving a formula for the flux across them; they go on to show that FTLE ridges are material lines to a very good degree of approximation.

It is suggested that we cite a wide body of literature regarding flow kinematics, atmospheric rivers and fronts; many of such studies were produced by Dr. Schultz's group and collaborators. We appreciate the quality and value of the contributions presented in these suggestions and will consider adding some of them in the revised manuscript. However, we do not aim to provide an extensive review on each of the concepts we explore: our aim is to introduce the FTLE and the concept of LCSs to the broad meteorology community as well as providing sufficient background literature to support the interpretation of our novel results. Moreover, most of the suggested literature is

around Eulerian metrics, such as the Okubo-Weiss criterion or the instantaneous Lyapunov exponent. While these are powerful diagnostics for instantaneous features or steady flows, they have limited ability to diagnose structures of tracer accumulation in unsteady flows. This is especially the case considering that moisture in the atmosphere has an average residence time of at least a few days, which is enough time for the moist parcels to explore large-scale turbulence and be shaped accordingly.

It is suggested that Figure 15 of Thomas and Schultz (2019) is, quoting Dr. Schultz, "very similar" to our Figure 6. Albeit somewhat related through the concept of airmass interface, the figures differ in more than one aspect: (1) the Atlantic ITCZ is not visible in their plot; (2) they capture a signal dominated by topography over South America. We believe that the difference between our results reflect that different methodologies were employed. The authors employed the asymptotic contraction rate, which is equivalent to the instantaneous local Lyapunov exponent, and, therefore, an Eulerian quantity. The authors also perform their analysis at the vertical level of 850 hPa. Our methodology employs a fully Lagrangian metric in a vertically integrated flow.

We appreciate the suggestions around the flow of the text and connection between chapters. These will be considered in the revised manuscript.

---

## Author Response (AR2)

**Reply to anonymous reviewers and list of changes**

**Gabriel M P Perez**

March 2021

**1 Response to Reviewer 1**

**1.1** General comments**

We appreciate the Reviewer's insightful comments and suggestions. The Reviewer pointed out the following general issues:

**1** - Connection between sections: We have added transition paragraphs at the end of each section to motivate the next. The text now reads more seamlessly.

**2** - Motivation for going to a climatological study: We have now made it more explicit in the introduction the need for running the LCS detection algorithm in a climatology. We have incorporated the Reviewer's suggestion to motivate it through the need of investigating the impact of LCSs in moisture and rainfall at different scales.

**3** - Conclusion improvements: We have expanded the conclusion section adding more on the potential applications of the methodology for convergence zones in other regions as well as forecast applications.

Other issues pointed in the Reviewer's general comment are addressed in the original submission. For example, the reviewer questions about the period employed in the climatological analysis. The period is stated in Line 161 of the original submission. Similarly, the Reviewer questions about how the methodology was applied in the entire climatology. This is expressed in the methodology diagram in Figure 3: the method is repeated in sliding time windows of 2 days separated by 6 hour intervals.

**1.2** Specific comments**

Line 22 We appreciate the Reviewer's suggestion to use "coherent winds" instead of "coherent trajectories". However, trajectories are used to highlight the Lagrangian nature of historical definitions of the ITCZ: it was considered to be and interface of air parcels originated from both hemispheres; thus the use of "coherent trajectories".

Line 23 We thank the Reviewer for the question. We consider that previous studies were focused on both quantitative and qualitative analyses. However, the automated methodologies employed by them required previous knowledge of the phenomena, such as the typical shape and intensity of the SACZ cloud band. Thus, heuristic rules were developed to identify SACZ events that attended their existing expectations.

**Line 130** We thank the Reviewer for this suggestion. The formulation of the Cauchy-Green tensor was done in 3D Cartesian space by transforming the latlon departure points in x, y, z coordinates. We have added this information in the revised version.

Line 169 : We thank the Reviewer for this suggestion. We have performed sensitivity tests to identify the smallest relaxation angle able to still capture LCSs associated with convergence zones attempting to preserve properties of Shadden's (2005) LCSs. Peikert and Sadlo (2008) suggest  $45^{\circ}$ , but we found that  $15^{\circ}$  sufficed for our case. We have included this brief explanation in the revised manuscript. We haven't performed further analysis regarding shear regions along the Andes, but we plan to do so in future studies.

Line 176: No, the +-20% sensitivity tests refer only to the intensity and length filters. We thank the r for this suggestion and included a short table summarising the parameters employed in the methodology.

**Figure 6** : As suggested by the Reviewer, we have now included a definition of "frequency of occurrence" in the manuscript.

**Line 202** : We appreciate the Reviewer's suggestion and have rephrased the definition of LCS accordingly.

Line 210-214 : We agree with the Reviewer that LCSs in the neighborhood of the Andes can be shear LCSs. However, we do not believe this to be the case of the structure labeled as "1" in Figure 5. This structure originates around a cyclonic circulation feature in South Atlantic and progresses with the confluence of the front associated with the cyclone. Furthermore, this event, as described in Line 191 of the original submission, was classified by Brazilian meteorology agencies to be an event of South Atlantic Convergence Zone.

We think that the source of confusion is that we positioned the label "1" in Figure 5 at an unfortunate location. We have replaced it to make it clear that we refer to the attractive LCS described by the Reviewer and we have attempted to clarify its interpretation in the text.

Line 241 : We thank the Reviewer for this comment and provide relevant citations as requested. However, we disagree with the Reviewer's comment

about the relationship between FTLE and mixing. The backwards FTLE at a given time is an integrated measure of the attraction of trajectories arriving in a neighborhood. Thus, ridges of the backwards FTLE can diagnose high mixing efficiency because arriving air parcels underwent substantial stretching. This is consistent with the concept of mixing proposed by Ottino (1989): "Mixing is stretching and folding".

**1.3** Technical comments**

We appreciate the Reviewer's suggestions about Figures 8 and 4. We have improved these figures accordingly as well as provided an improved color palette in Figure 1.

**1.4 References**

- Ottino, Julio M., and J. M. Ottino. The kinematics of mixing: stretching, chaos, and transport. Vol. 3. Cambridge university press, 1989.
- Peikert, Ronald, and Filip Sadlo. "Height ridge computation and filtering for visualization." 2008 IEEE Pacific Visualization Symposium. IEEE, 2008.
- Shadden, Shawn C., Francois Lekien, and Jerrold E. Marsden. "Definition and properties of Lagrangian coherent structures from finite-time Lyapunov exponents in two-dimensional aperiodic flows." Physica D: Nonlinear Phenomena 212.3-4 (2005): 271-304.

**2 Reviewer 2**

**2.1 General comments**

We appreciate and thank the Reviewer for the insightful comments and suggestions. The Reviewer pointed out the following general issues:

**1** - **Abbreviations** We have addressed the issue of unexplained abbreviations in the revised manuscript.

**2** - **Title** We have considered the Reviewer's suggestion about including "Coherent structures" in the title. We will be adding these as keywords for the final manuscript. However, our analysis is not limited to coherent structures (FTLE ridges). It also includes correlations between a metric of mixing efficiency (the FTLE scalar field) and water vapour and rainfall. Since "coherent structures" can be studied within the framework of chaotic mixing, we believe that the current title better express the contents of the paper.

**2.2 Minor comments**

Line 45 We thank the Reviewer for this suggestion. We have detailed the definition of convergence zone as suggested.

Further approaches do not count elongated structures as coherent sets, why does this work here? We thank the Reviewer for this comment. However, we are not entirely sure about which part of the manuscript the Reviewer is referring to here. We do describe the LCSs on our case study as a "coherent ensemble" because they last for a considerable amount of time and seem to move as a group. To avoid confusion we have removed the word "coherent" from this sentence (Line 221).

"Are there further approaches to identify coherent structures of meteorological phenomena besides the FTLE?" We thank the Reviewer for this question. In meteorology there are many methods to identify coherent structures such as hurricanes and extratropical cyclones. However, from a purely kinematical point of view, there is a limited but growing number of approaches, such as the geodesic transport barriers of Haller and Beron-Vera (2012).

**Fig 4** Here the departure positions are shown in units of degrees Latitude and Longitude. We have now clarified this in the figure caption.

**TCWF** We thank the reviewer for this observation. We have specified this abbreviation in the revised version.

"figures are often placed far away in the paper" We thank the reviewer for this comment. We have rearranged the figures in the revised version.

Figure 5 We appreciate the Reviewer's comment and have added a brief explanation of the LCS computation. However, this computation is described in detail in the Methodology section.

Are  $x_0$  and  $x_1$  in  $\mathbb{R}^2$ ? Yes,  $V_{\rho_v}$  is vertically integrated such that  $V_{\rho_v} = V_{\rho_v}(x, y)$  where x, y are points in  $\mathbb{R}^2$  as they correspond to locations in the Earth's surface. Thus, the points  $x_0$  and  $x_1$  are in  $\mathbb{R}^2$ . We have clarified this in the revised version.

**Page 8 "How do you identify the trajectory"** Here we integrate trajectories for every grid point available in ERA5 grid. So there is no prior choice of trajectory. This have been clarified in the revised version.

"Why is the rate sigma exponential and not logarithmic" Lyapunov exponents are typically defined exponential rates connecting the separation of trajectories at two different times. In the case of the backwards FTLE, we formulate that the separation  $\delta$  at the departure time  $t = t_0$  is equal to the the separation at the arrival time  $t = t_1$  multiplied by an exponential term:

$$\delta(t_0) = \delta(t_1) e^{\sigma \Delta t} \tag{1}$$

The separation ratio  $\delta(t_0)/\delta(t_1)$  is expressed by the largest eigenvalue of the Cauchy Green tensor. Thus, Equation 4 in the original submission can be obtained by applying the logarithm in the equation above and isolating the exponential rate  $\sigma$ , i.e., the FTLE.

Why is  $\lambda_{max}$  at the same time an eigenvalue and a norm? We thank the reviewer for this comment.  $\lambda_{max}$  is not a norm and we have fixed this in the revised manuscript.

**Page 8: time resolution** 2 days is the length of the time window, the time resolution employed to compute the trajectories was 6 hours. We have clarified this in the revised version.

Line 160 We thank the reviewer for this suggestion. We have referred to Eq. 1 in the revised version.

How does this method handle sources and sinks The flow  $V_{\rho_v}$  used to calculate the trajectories should not be affected by the total amount of water in the column as the water vapour density  $\rho_v$  is both in the numerator and denominator of Eq. 1, yielding a flow that is not dependent on the horizontal water vapour distribution. The role of  $\rho_v$  is only to provide a vertical scaling that favours moist levels. Therefore, we hope that sources and sinks do not affect our detection of LCSs. We have added this discussion on the revised manuscript.

What is the absolute number of events? The absolute number of events would depend on the size of the boxes and the time interval in which the events are accounted for. The fine grid we considered in this work results in the low fraction of events per box shown in Figure 6. A first order guess of the absolute number of days in which an event occurred would be something between 0 and 50 days per year, depending on the region of the domain.

Motivate the correlations in 6.2 We thank the reviewer for this comment and have included a motivating sentence as suggested.

Figure 8 : We have enhanced the size of the VIMF vectors so they are clearer.

Lines 303-304 We have added references as requested.

Lines **311-312** We have added references as requested.

**3** Response to Short Comment 1**

We thank Dr. David Schultz for his insightful comments and for his inerest in our study. Here we will be addressing each of the three parts of his comments.

**3.1 Part A - Literature that should be cited**

It is suggested that we cite a wide body of literature regarding flow kinematics, atmospheric rivers and fronts; many of such studies were produced by Dr. Schultz's group and collaborators. We appreciate the quality and value of the contributions presented in these suggestions and will considering adding some of them in the revised manuscript. However, we do not aim to provide an extensive review on each of the concepts we explore: our aim is to introduce the FTLE and the concept of LCSs to the broad meteorology community as well as providing sufficient background literature to support the interpretation of our novel results. Moreover, most of the suggested literature is around Eulerian metrics, such as the Okubo-Weiss criterion or the instantaneous Lyapunov exponent. While these are powerful diagnostics for instantaneous features or steady flows, they have limited ability to diagnose structures of tracer accumulation in unsteady flows. This is especially the case considering that moisture in the atmosphere has an average residence time of at least a few days, which is enough time for the moist parcels to explore large-scale turbulence and be shaped accordingly.

It is suggested that Figure 15 of Thomas and Schultz (2019) is, quoting Dr. Schultz, "very similar" to our Figure 6. Albeit somewhat related through the concept of airmass interface, the figures differ in more than one aspect: (1) the Atlantic ITCZ is not visible in their plot; (2) they capture a signal dominated by topography over South America. We believe that the difference between our results reflect that different methodologies were employed. The authors employed the asymptotic contraction rate, which is equivalent to the instantaneous local Lyapunov exponent, and, therefore, an Eulerian quantity. The authors also perform their analysis at the vertical level of 850 hPa. Our methodology employs a fully Lagrangian metric in a vertically integrated flow.

**3.2 Part B - Consistency of the manuscript with previous literature**

Here it is suggested that some abbreviations that we have employed in the original manuscript are not in accordance with what is usually employed in the moisture transport literature. These abbreviations have now been spelled out fully, also complying with the Editor's comments.

It is also suggested that we provide a reference about the derivation and mathematical notations of the strain tensor. In this study we have adopted the notations presented in Haller (2015), which is a comprehensive review about Lagrangian coherent structures. We have accepted the suggestion and added this reference in Section 2.

The commentator also states that Cohen and Kreitzberg (1997) employed a 12 hour timescale to identify transport barriers and asks for our rationale for a 2-day time scale. Our rationale is described in the last paragraph of Section 2.2 of the original submission: we used a timescale that is long enough to explore large-scale structures and not be influenced by diurnal circulations. But it is not too long as to filer out the effect of extratropical cyclones, as we are interested in this type of structure. Moreover, as we stated in the original submission, we tested a range of timescales (1, 3 and 4 days) and obtained robust results.

**3.3 Part C - Terminology and Readability**

Here the commentator points to aspects related to the readability of the manuscript. Some of these, such as reducing the number of acronyms and improve the connectivity between chapters, have been addressed in the revised version, also complying with the reviewers' comments. We have also improved the discussion in Section 7, especially regarding potential applications of the method.

The commentator criticises the title of our last section: "Summary and Conclusions" by suggesting that the two words are the same. Here we point to the dictionary definition of summary: "a compendium of previously stated facts or statements" (https://www.dictionary.com/browse/summary). In the final section we aim to repeat the discussion in a more concise way as well as presenting the main conclusions, hence "Summary and Conclusions".

The commentator suggests that we use the word "automated" instead of "objective" to describe our methodology. We disagree with this. We suggest the reading of the seminal work of Shadden et al. (2008) on Lagrangian Coherent Structures (LCSs). The authors demonstrate the objectiveness of FTLE ridges by deriving a formula for the flux across them; they go on to show that FTLE ridges are material lines to a very good degree of approximation.

**4 Response to editor comments**

We thank the editor for his comments. To improve readability by reducing the number of abbreviations. The following abbreviations were removed in the revised version: SASH, TCWF, VIMF.

**5 List of relevant changes in manuscript by order of appearance in the document**

The following items are the relevant changes in the revised manuscript, according to the Reviewer's suggestion. The figures were rearranged to be closer to where they are cited in text as requested by a reviewer. The major improvements are discussions on the justification of a climate study and the potential applications of the methodology.

1. Section 1.1: enhanced discussion of mixing and coherent structures

2. Section 1.2:

Improved color palette for Figure 1

Justification for the need of a long-term study

3. Section 2.1:

Added discussion about sources and sinks

Added sentence in the final paragraph to connect with the following section

4. Section 2.2: Added details about the Cartesian and spherical coordinates Referencing Haller 2015 about the derivation of the strain tensor and FTLE metrics.

5. Section 3:

Included a table summarising the relevant parameters of the methodology

Added a short sentence about the choice of tolerance angle for the ridge detection

- 6. Section 4: Added a short sentence in the final paragraph to introduce the next section
- 7. Section 5: Added a paragraph in the end to justify the need to perform a climatological study and to introduce the next section
- 8. Section 6.1:

Included a definition of frequency of occurrence

Included a final paragraph to justify the next section

9. Section 6.2:

Included a reference about correlating FTLE with atmospheric variables

Included a final paragraph to justify the next section

10. Section 6.3:

Renamed the section

Added introduction to the next section in the last paragraph

11. Section 6.4:

Renamed the section

Enhanced the discussion on the connection between Rossby waves and large-scale mixing

12. Section 7:

Expanded the discussion about the application of the method on other locations and its use in operational weather forecasting.